# Simulating Soil-Plant-Climate Interactions and Greenhouse Gas Exchange in Boreal Grasslands Using the DNDC Model

Daniel Forster [1],*[ID], Jia Deng [2][ID], Matthew Tom Harrison [3][ID] and Narasinha Shurpali [1],*[ID]

1    Natural Resources Institute Finland (Luke), Halolantie 31 A, 71750 Maaninka, Finland
2    Earth Systems Research Center, Institute for the Study of Earth, Oceans and Space, University of New Hampshire, 39 College Road, Durham, NH 03824, USA
3    Tasmanian Institute of Agriculture, University of Tasmania, Newnham Drive, Launceston, TAS 7248, Australia
*    Correspondence: danforster.grassland@hotmail.com (D.F.); narasinha.shurpali@luke.fi (N.S.)

**Abstract:** With global warming, arable land in boreal regions is tending to expand into high latitude regions in the northern hemisphere. This entails certain risks; such that inappropriate management could result in previously stable carbon sinks becoming sources. Agroecological models are an important tool for assessing the sustainability of long-term management, yet applications of such models in boreal zones are scarce. We collated eddy-covariance, soil climate and biomass data to evaluate the simulation of GHG emissions from grassland in eastern Finland using the process-based model DNDC. We simulated gross primary production (GPP), net ecosystem exchange (NEE) and ecosystem respiration (Reco) with fair performance. Soil climate, soil temperature and soil moisture at 5 cm were excellent, and soil moisture at 20 cm was good. However, the model overestimated NEE and Reco following crop termination and tillage events. These results indicate that DNDC can satisfactorily simulate GHG fluxes in a boreal grassland setting, but further work is needed, particularly in simulated second biomass cuts, the (>20 cm) soil layers and model response to management transitions between crop types, cultivation, and land use change.

**Keywords:** ecophysiological modelling; boreal agriculture; greenhouse gases; model evaluation; DNDC; soil organic carbon; net-zero

## 1. Introduction

Global temperature rise is placing increased pressure on boreal lands from agricultural land-use expansion and intensification to meet the needs of a burgeoning population [1]. Boreal regions also present significant opportunities for greenhouse gas (GHG) mitigation and may have the potential to act as further sinks of atmospheric carbon [2]. United nations sustainable development goals [3] have identified sustainable agriculture (goal 2) and protection of ecosystems (goal 15) as key components of the overarching strategy to address climate change. We are thus at a critical juncture in time in which holistic assessments and planning decisions regarding management and land-use trade-offs will be key to ensuring long-term sustainability [4]. Given the risks and opportunities afforded by climate change in boreal areas, decision making needs to balance the manifold factors involved to balance increasing societal requirements for the long-term preservation of the natural capital on which our agri-food systems depend [5–8].

Agroecological models have increasingly been used to simulate the effects of management [9,10] and land use conversion [11] on biomass production [12–14], canopy-level physiology [7,8], GHG emissions [15–18], profitability [19] and soil carbon/nitrogen cycling [20]. When properly calibrated, such models have been able to accurately simulate these agriculturally and environmentally important variables and thus have the potential to assist in management planning while promoting both production and environmental protection, including climate change mitigation by reducing GHG emissions [21–23].

The DeNitrification-DeComposition (DNDC) model [24] has been long in use worldwide, and has demonstrated sufficient accuracy in modelling crop growth and GHG emissions in cooler regions [11,25–33]. The extensive use of this model in colder climates and its successful simulation of relevant outputs would make it seem a sensible choice for application in Scandinavia, although significant uncertainty remains around the performance of DNDC in the challenging boreal environment due to the lack of studies measuring GHG and other variables for use in model calibration in regions with extensive durations of sub-zero temperatures [34].

Of the few agroecological modelling studies that have been conducted in boreal zones, results for DNDC have been promising. For example a study by He et al. [31] modelled the effects of manuring on $N_2O$ emissions in Canadian grasslands, evaluating against measured data and while most metrics were graded "good" to "fair", soil water and N simulations were only "acceptable", and the authors also recommended improvements to the models soil freeze-thaw simulations, as well as soil microbial and water processes in grasslands. Abdalla et al. [27] used DNDC to evaluate soil respiration in the Republic of Ireland from grassland and conventionally managed arable fields under three climate scenarios: a baseline of measured climate data and both high and low temperature sensitivity scenarios. They indicated that DNDC could effectively model soil respiration in both pasture and arable, underestimating annual $CO_2$ efflux by only 13% and 8% respectively. Another study in the Republic of Ireland [35] examined management effects on $N_2O$ emissions from grasslands using a two year (2008–2009) dataset. The study showed that flux estimates tended to be higher than those estimated using IPCC emissions factors, and the authors suggested that soil parameters needed further calibration for optimum performance. A study in Northern Ireland [36] used DNDC95 to evaluate SOC density and annual changes in temperate long-term grassland soils. They found that the model underestimated SOC by 0.9 t C ha$^{-1}$, yr$^{-1}$, a difference which was explained by differences in supplied N and differences in soil C, rainfall, and air temperature as well as soil physiochemical variables. Because most of the studies with DNDC have been conducted in temperate or cool temperate regions with more moderate temperatures, different soils and different land management, such results may not translate to Scandinavian conditions, hence the need for a DNDC study in Finland.

The purpose of this study was to evaluate DNDC performance in simulating soil microclimate, biomass production and GHG fluxes against eddy-covariance measured NEE, here defined as the net exchange of $CO_2$ between the ecosystem and atmosphere in kg C ha$^{-1}$, and associated soil and plant data in a legume grassland in eastern Finland with attention to model accuracy in simulating freeze-thaw cycles.

## 2. Materials and Methods

### 2.1. Site Description

This study was conducted at the Antilla field site, located in Maaninka, eastern Finland, (63°09′ N, 27°140′ E, 89 m a.s.l.); a location with mean annual temperature (1981–2010) of 3.2 °C and mean annual precipitation of 612 mm year$^{-1}$. In the study period, mean, maximum and minimum temperatures were 5.1, 25.0 and –26.5 °C, respectively, and annual rainfall was 613 mm, 515 mm, and 532 mm for 2017, 2018 and 2019, respectively (Figure 1). The study site was a 6.3 ha agricultural field where the mineral soil is classified as a haplic cambisol (silt loam: clay 25 ± 7.8%, silt 53 ± 9%, sand 22 ± 7.8%) based on the USDA classification system (Table 1).

The field was cultivated with a mix of timothy (*Phleum pratense* L. cv Nuuti), meadow fescue (*Festuca pratensis*) and red clover (*Trifolium pratense*) at a rate of 15 and 5 kg ha$^{-1}$ for grasses and legumes respectively in 2015, was reseeded in May 2017, and ploughed in the autumn of 2018 when glyphosate was also applied. The grassland was renewed in Spring 2019 with a cover crop of barley (*Hordeum vulgare* L.) in a fresh rotation. Mineral fertilizer was applied (106 kg N, 28 kg P and 50 kg K/ha$^{-1}$) divided evenly over two applications at the start of the growing season and after first cut in 2017 and 2018, with a single application

in the renewal year of 45 kg N, 20 kg P, 38 kg K/ha$^{-1}$. Cuts were carried out twice annually in late June and mid-August for 2017 and 2018, and a single cut in early August in the renewal year with a disc mower to remove biomass material to 8 cm, which was then swathed, baled, and removed from the field site.

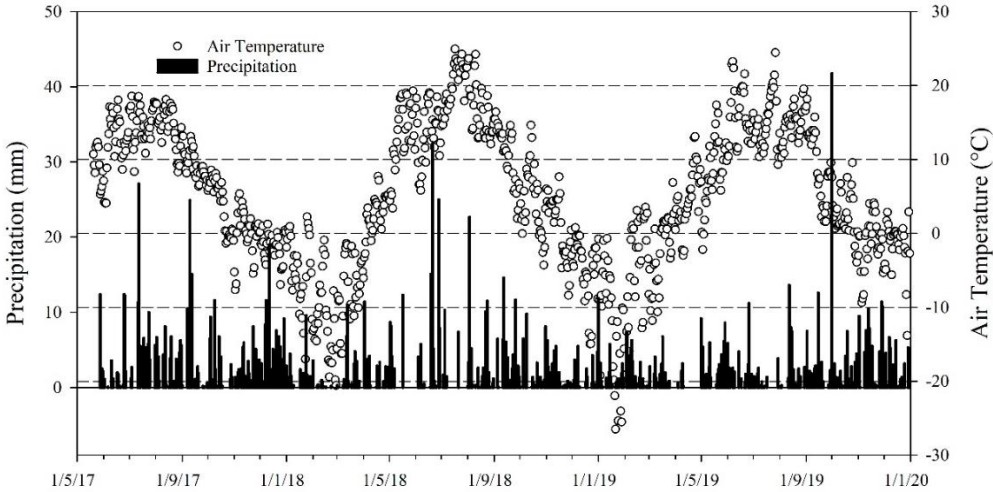

**Figure 1.** Measured air temperature (°C) and precipitation (mm) per day over the modelled period (Data obtained from Finnish Meteorological Institute (FMI).

**Table 1.** Topsoil (0–5cm) measurements in the Antilla site.

| Unit | Value |
| --- | --- |
| Soil pH | 5.8 ± 0.19 |
| EC mS m$^{-1}$ | 14 ± 2.4 |
| SOM% | 5.2 ± 0.9 |
| SOC% | 3.0 ± 0.52 |
| C/N ratio | 15 ± 0.4 |
| Total N% | 0.2 ± 0.03 |
| P mg L$^{-1}$ | 5.4 ± 1.28 |
| K mg L$^{-1}$ | 104 ± 12.9 |

*2.2. Eddy Covariance Data*

An eddy covariance (EC) tower was setup in the centre of the study area in 2017. The $CO_2$ and $H_2O$ measurements were performed by a closed-path EC system with adjacent weather station providing supporting climate and meteorological data. The EC system was a Li-7000 infrared gas analyser (IRGA, for $CO_2$ and $H_2O$ mixing ratios, Li-COR inc., Lincoln, NE, USA), a sonic anemometer (wind velocity, sensible heat flux and sonic temperature components R3-50, Gill Instruments Ltd., Lymington, UK) mounted 2.5 m above the soil surface on an instrument tower. Air samples pass through a heated intake tube at a flow rate of 10 L min$^{-1}$, (a PTFE tube, internal diameter 6 mm, length 8 m) with two 1.0 µm pore size filters (Gelman$^{®}$). The IRGA was housed in a climate-controlled cabin and calibrated monthly during the growing season. Supporting climate data were net radiation, relative humidity, photosynthetically active radiation, soil temperature, volumetric soil water content (at 5 and 20 cm depths) and air pressure. A CR3000 (Campbell Scientific Inc., Logan, UT, USA) 10 Hz data logger collected raw EC data. Missing supporting meteorological data were gap filled using data from the Maaninka weather station (Finnish Meteorological Institute), located 6 km southeast of the field site. Eddy covariance data processing was carried out using EddyUH [37]. Annual EC data for the period of the experiment (20 May 2017 to 31 May 2020) were employed for calibration and evaluation purposes.

### 2.3. The DNDC Model

The DeNitrification-DeComposition (DNDC) model is a process-based biogeochemical model developed for quantifying C sequestration as well as emissions of C and N gases from agricultural ecosystems [38–40]. The model is comprised of six sub-models: soil climate, plant growth, decomposition, nitrification, denitrification, and fermentation. The soil climate, plant growth, and decomposition sub-models convert the primary model drivers, such as climate, soil properties, vegetation, and anthropogenic activity, into soil environmental factors (e.g., soil temperature and moisture, pH, redox potential) and concentrations of substrates of relevant biogeochemical processes. The nitrification, denitrification, and fermentation sub-models simulate C and N transformations that are mediated by soil microbes and controlled by soil environmental factors and concentrations of relevant substrates [40,41]. The DNDC model adopted in this study was further improved to simulate surface energy exchange, soil frost and thaw dynamics, and C gas fluxes in cold regions [42–44].

The DNDC model requires daily climate data, including minimum and maximum temperatures (°C) and rainfall (mm), as well as humidity (%), windspeed (m/s), and solar radiation (MJ/m$^2$). The model also requires rainfall N concentrations, atmospheric $NH_3$ and $CO_2$ concentrations, and annual increases of atmospheric $CO_2$, although model default values can be used when such information is not available. In addition, we used measurements of actual snow depth to drive the model [43]. Using measured snow depth in this way improves the surface energy balance and hence the simulated soil temperature and water filled pore space (WFPS cm$^3$/cm$^3$). Soil input data include soil properties according to the USDA soil classification system, as well as information on initial soil organic carbon (SOC), pH and other soil physiochemical factors. The input parameters of farming management practices, including crop types, planting and harvest dates, tillage, fertilization, residue return, and irrigation, were taken from Li et al. (*unpublished*). The Antilla site using soil data from Lind et al. [45]. Initial model output evaluations were carried out against EC data for net ecosystem exchange (NEE), gross primary production (GPP, defined as the total $CO_2$ taken up by the ecosystem in photosynthesis in kg C ha$^{-1}$) and ecosystem respiration (Reco, defined here as the total ecosystem respiration (sum of aboveground plant and root (autotrophic) and heterotrophic respiration) in kg C ha$^{-1}$). Output data assessment and analysis was conducted in R version 4.1.1 (R development core team 2021) and RStudio (version 1.4.1106).

### 2.4. Model Initialisation Calibration and Evaluation

The model was calibrated using data in 2017 and 2018 and then ran continuously from 2017 to 2019, with 2019 used to evaluate the model. The calibration process included optimisation of crop phenological parameters (thermal days to maturity, biomass fraction, root: shoot ratios, stem: leaf: grain fraction, water demand, N fixation index, and optimum temperature, Table 2). For modelling purposes, a 16-year spin-up period was introduced to allow time for simulated soil carbon stocks to stabilise.

The crop setup consisted of two systems, the first consisted of a perennial grass ley (land put down to grass and/or clover for a limited period) calibrated to simulate a grass/legume mixture. The second system simulated the same, with the addition of a barley cover crop, though no cuts were carried out in 2020 (Table 3). Input parameters for soil and management are as described in Section 2.1, though fertiliser applied (106 kg N/ha/year) was divided equally among the number of fertilisation events for that year.

Both calibration and evaluation were conducted using base R and the package 'Hydro-GOF' [46]. Four evaluation metrics were used to evaluate model performance against measured GHG fluxes. These were Spearman's rho ($\rho$, Equation (1)), where $\rho$ is the Spearman's rank correlation coefficient, $d_i$ is the difference between the two ranks of individual measured and corresponding simulated data pairs, and $n$ is the number of observations. Mean absolute error (MAE, Equation (2)) which assesses the size of prediction errors at the individual level, but does not allow comparison between positive and negative predictors. Root mean square error (RMSE, Equation (3)) measures absolute quadratic prediction error.

$S_i$ is the simulated, and $M_i$ are the measured variables. Percent bias (*pBIAS*, Equation (4)) gives a relative bias estimation to determine over or underestimation in the simulation.

**Table 2.** Crop calibration parameters used in this study. Figures in grey are automatically produced model outputs in response to calibration, and not directly subject to manipulation.

| Perennial Grass | Grain | Leaf | Stem | Root |
|---|---|---|---|---|
| Max. biomass production (kg C/ha/yr) | 400 | | | |
| Biomass fraction | 0.04 | 0.28 | 0.28 | 0.4 |
| Biomass C/N ratio | 35 | | | |
| Annual N demand (kg N/ha/yr) | 143 | | | |
| Thermal degree days to maturity | 1500 | | | |
| Water demand (g water/g dry matter (DM)) | 150 | | | |
| N fixation index (crop N/N from soil) | 1.5 | | | |
| Optimum temperature (°C) | 18 | | | |
| **Barley** | | | | |
| Max. biomass production (kg C/ha/yr) | 2496 | | | |
| Biomass fraction | 0.3 | 0.23 | 0.23 | 0.23 |
| Biomass C/N ratio | 45 | 75 | 75 | 85 |
| Annual N demand (kg N/ha/yr) | 129 | | | |
| Thermal degree days to maturity | 1500 | | | |
| Water demand (g water/g DM) | 150 | | | |
| N fixation index (crop N/N from soil) | 1 | | | |
| Optimum temperature for crop growth (°C) | 18 | | | |

**Table 3.** Management details used in DNDC simulations in the present paper based on Li et al. (*unpublished*). 'Model setup' indicates the division between calibration and evaluation datasets. The full model was run until May 2020, although that year was not used in model evaluation and is shown in grey to reflect this.

| Year | 1 (2017) | 2 (2018) | 3 (2019) | 4 (2020) |
|---|---|---|---|---|
| Model setup | Calibration dataset | | Evaluation dataset | NA |
| Crop: perennial grass | 18 May 2017–30 October 2018 | | 4 June 2019–31 May 2020 | |
| Cover crop: barley | NA | NA | 4 June 2019–31 May 2020 | |
| Cuts | 29 June | 26 June | 6 August | - |
| | 16 August | 7 August | | |
| Overseeding/reseeding | 18 May | NA | 16 August | - |
| Fertilisation | 22 May & 3 July | 22 May & 2 July | 2 July | * 22 May & 2 July |
| Tillage | NA | 30 September (crop killing till), 30 October | 3 June | - |

$$\rho = 1 - \frac{6 \sum d_i^2}{n(n^2 - 1)} \tag{1}$$

$$MAE = \frac{\sum_{i=1}^{n} |S_i - M_i|}{n} \tag{2}$$

$$RMSE = \sqrt{\frac{\sum_{i=1}^{n} (S_i - M_i)^2}{(n)}} \tag{3}$$

$$pBIAS = 100 \frac{\sum_{i=1}^{n} (S_i - M_i)}{\sum_{i=1}^{n} M_i} \tag{4}$$

Following setup and calibration we again ran the model continuously from 2000 to 2020 and used 2019 for model evaluation against gross primary production (GPP kg C ha$^{-1}$), net ecosystem exchange (NEE kg C ha$^{-1}$) and ecosystem respiration (Reco kg C ha$^{-1}$), soil temperature (°C) at 5 cm, and soil moisture (WFPS cm$^3$/cm$^3$) at 5 cm and 20 cm according to a pre-determined criteria (Table 4).

**Table 4.** Model evaluation based on classifications as per [27,31,34]. Overall scores are calculated as the mean performance of variables across all four metrics.

| Evaluation Method | Poor | Fair | Good | Excellent |
|---|---|---|---|---|
| Spearman's rank correlation ($\rho$) | 0.30 | 0.50 | 0.70 | 1.00 |
| MAE | 4.0+ | 3.0–3.9 | 2.0–2.9 | 1.0–1.9 |
| RMSE | $\geq$40 | 20–39 | 10–19 | 0–10 |
| pBias% | >20% | 15–20% | 11–15% | <10% |

## 3. Results

Greenhouse gas exchange (Figure 2), soil climate simulations (Figure 3) and crop biomass simulations (Figure 4) were generally in agreement with measurements through most of the modelled timeframe. For GPP, there was a good correlation $\rho$ between simulated and measured data, though MAE was poor, RMSE was fair, and the model pBias% underestimated and scored poor. For NEE, there was a good $\rho$ correlation between simulated and measured data, though MAE was poor, RMSE was fair, and the model scored good overall, with a small pBias% underestimate. For Reco, there was a good $\rho$ correlation between simulated and measured data, though MAE was poor, RMSE was good, but model pBias% underestimated and scored poor (Table 5).

**Table 5.** Evaluation results for simulated versus measured GHG exchange and soil microclimate. The 'Mean score' column represents the overall assessment when all four measures are accounted for.

| | $\rho$ | MAE | RMSE | pBias% | Mean Score |
|---|---|---|---|---|---|
| GPP (kg C ha$^{-1}$) | 0.80 ($p < 0.001$) | 21.3 | 35.1 | –20.7% | Fair |
| NEE (kg C ha$^{-1}$) | 0.72 ($p < 0.001$) | 16.6 | 26.7 | –14.2% | Fair |
| Reco (kg C ha$^{-1}$) | 0.85 ($p < 0.001$) | 10.9 | 14.2 | –22.8% | Fair |
| Soil Temp (°C) | 1.00 ($p < 0.001$) | 0.1 | 1.2 | 18.2% | Excellent |
| WFPS (cm$^3$/cm$^3$) 5 cm | 0.73 ($p < 0.001$) | 0.1 | 0.1 | –11.2% | Excellent |
| WFPS (cm$^3$/cm$^3$) 20 cm | 0.25 ($p < 0.001$) | 0.1 | 0.1 | –5.0% | Good |

For soil temperature there was a correlation $\rho$ score of excellent, MAE was excellent, RMSE was excellent and pBias% scored fair, and showed that the model overestimated compared to measured data. Soil water (WFPS) at 5 cm scored "good" $\rho$ for Spearman's correlation, MAE was excellent, and RMSE was also excellent, while there was a small, but good underestimation for pBias%. For WFPS at 20 cm there was a poor $\rho$ correlation, whereas MAE was excellent and RMSE was excellent, whilst there was only a small pBias% underestimation, which scored excellent (Table 5).

The DNDC model simulated seasonal patterns of GHG exchange (Figure 2) and soil climate (Figure 3) well. Seasonally, GHG's tended to be close to 0 between October and April, although in the evaluation year eddy covariance did not show the simulated uptick in ecosystem respiration and NEE (Figure 2b,c) until six weeks later in mid-June, and DNDC did not pick up on this and also did not pick up on wintertime ecosystem respiration (Figure 2c).

Soil temperature at 5cm (Figure 3a) was very similar to measured data, although WFPS at 5 cm (Figure 3b) showed less accuracy in January and May, and in the 20 cm layer DNDC overestimated in May 2018 and underestimated between December 2018 and May 2019 (Figure 3c).

Biomass (DM kg/ha$^{-1}$) simulation were compared with measured cuts using an independent sample $t$-test. Measured DM (mean = 3117, sd = 915) was compared to simulated DM (mean = 4517, sd = 926) and there was a significant difference between the two $t$ (8) = −2.4, $p < 0.05$.

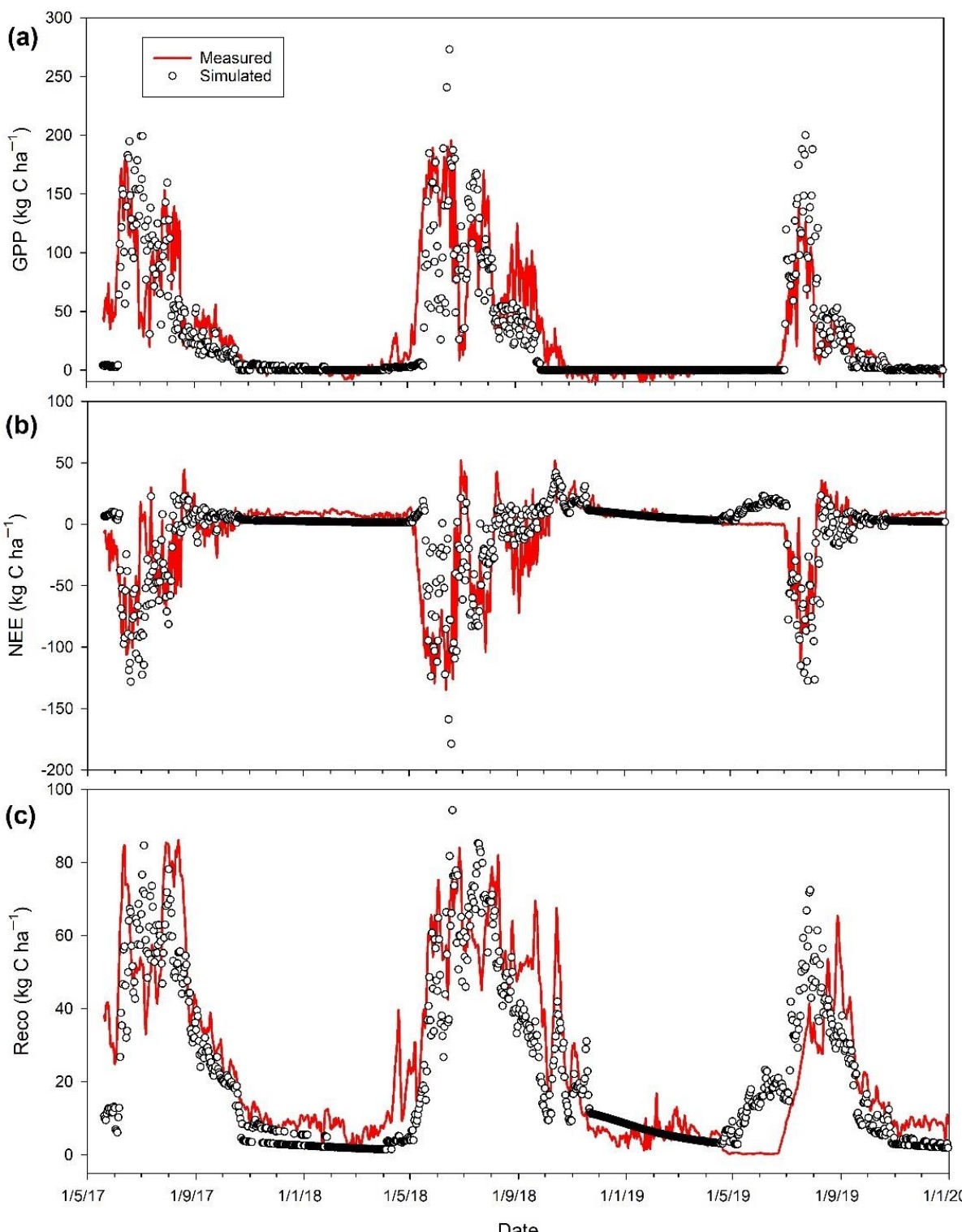

**Figure 2.** Simulated and measured GPP (**a**), NEE (**b**) and Reco (**c**) over the three modelled growing seasons.

Total annual GHG exchange was measured for the calibration and evaluation years 2018 and 2019 (Table 6) indicating that DNDC simulated GPP, NEE and Reco followed a similar pattern to measured eddy-covariance figures and both field measurement and simulations showed reduced respiration in 2019 compared to 2018.

The DNDC model predicted that the Antilla site would be an overall sink of atmospheric C by an average of –1.17 T C ha$^{-1}$ yr$^{-1}$, although eddy covariance indicated that in 2019 the field site was a small source of C, but still an overall sink.

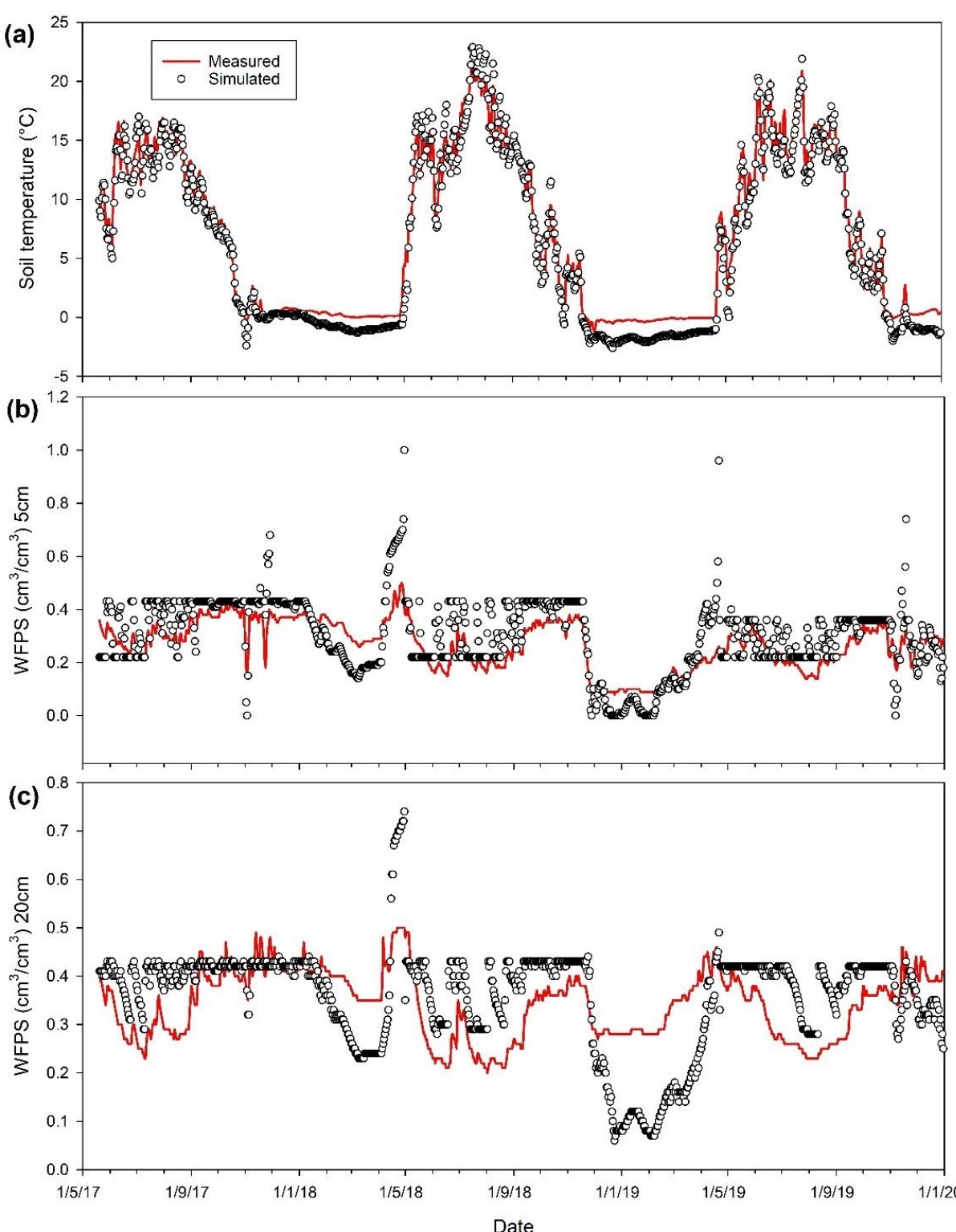

**Figure 3.** Comparison of soil temperature (°C) at (**a**) 5 cm, WFPS at (**b**) 5 cm, and (**c**) WFPS at 20 cm.

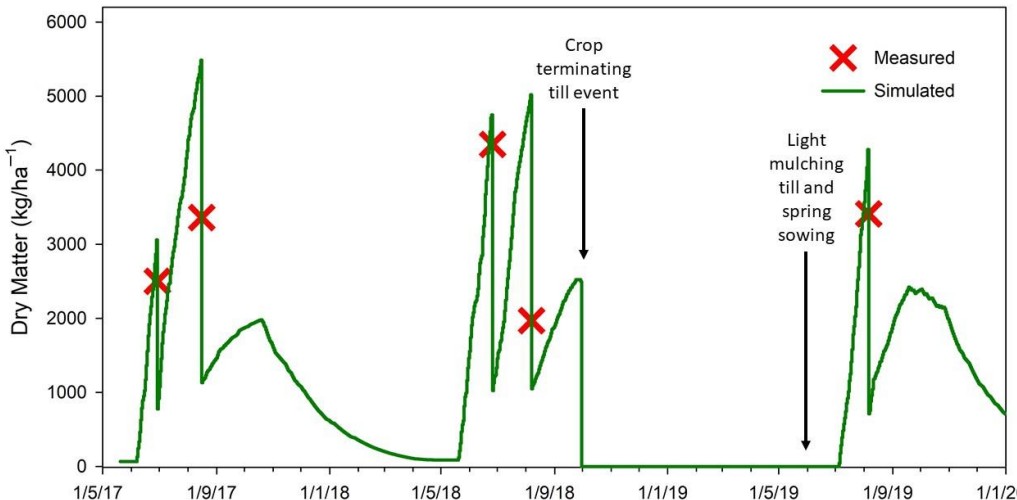

**Figure 4.** Comparison of simulated (—) and measured (×) biomass production (DM kg/ha$^{-1}$) for growing seasons 2017 to 2019. For 2017, the difference for cut 1 was 5.6% and cut 2 was 21.3%. For 2018 cut 1 was 4.0% and cut 2 was 30.5%, and for 2019 the simulated cut was 8.7% higher than measured DM and the black arrows ('ploughing event' and 'tillage and spring sowing').

**Table 6.** Comparison of total annual GHG exchanges for calibration (2018) and evaluation (2019) years. The 2017 year is omitted as measurement did not begin until May.

| Calibration/Evaluation Years | Measured | Simulated | Difference |
|---|---|---|---|
| GPP (T C ha$^{-1}$ yr$^{-1}$) | | | |
| 2018 | 14.79 | 10.09 | 4.70 |
| 2019 | 4.15 | 5.19 | 1.04 |
| Mean | 9.47 | 7.64 | −1.83 |
| NEE (T C ha$^{-1}$ yr$^{-1}$) | | | |
| 2018 | −3.64 | −1.97 | 1.68 |
| 2019 | 0.22 | −0.35 | −0.57 |
| Mean | −1.71 | −1.17 | 0.55 |
| Reco (T C ha$^{-1}$ yr$^{-1}$) | | | |
| 2018 | 11.15 | 8.12 | −3.03 |
| 2019 | 4.58 | 4.84 | 0.26 |
| Mean | 7.87 | 6.48 | −1.38 |

## 4. Discussion

This study begins to address the dearth of model evaluations for boreal managed grasslands by presenting a comparison of model simulations for DNDC with eddy-covariance GHG flux, soil climate and biomass dry-matter production data. There are relatively few papers containing model evaluations for grasslands in boreal areas, but since model testing during their creation is necessarily limited to regions available to the model creators, it does not follow that they are applicable elsewhere and a careful assessment of regional conditions is necessary to understand how a model might behave in novel environments.

A number of papers have assessed the DNDC model in cool weather regions in Canada [29,31] and Northern Europe [27,43,44], making it an ideal candidate for evaluation with a view to using in the boreal agricultural landscapes of Scandinavia. Furthermore, the present study adds to this by demonstrating that the model can produce fair estimations of the key GHG fluxes and good estimations of soil climate conditions when compared to measured field data, but that there remain a number of uncertainties that would benefit from further elucidation, for example crop parameterisation and root processes, soil moisture simulations and GHG responses to changes in management.

*4.1. GHG Exchange*

In general, the DNDC model simulated GHG exchange fairly well, there was a tendency to underestimate GPP and Reco compared to measured data. This may be linked to the model simulating little or no respiration during the colder months, and to both increase more slowly and decrease more rapidly at the commencement and ending of the growing season compared to measured data. Using the improved DNDC model [42] improved model performance which was able to simulate GHG satisfactorily.

In the evaluation dataset, spikes of GPP and Reco in August 2018 and 2018 were probably related to increased rainfall noted in those periods (Figure 1), and a simulated increase in NEE and Reco beginning in May 2019 was not matched by observed data (Figure 2b,c). This may reflect a modelled increased soil respiration since the model simulated a 'crop terminating till' in October 2018, followed by a 'light mulching till' and seeding on 3 and 4 June respectively (Table 3). The residue incorporation following crop termination and tillage event increased the modelled soil respiration. However, they simulated an increase in soil respiration before crop germination in contrast to the findings of Oertel et al. [47] who found that bare soils tended to have lower GHG flux than other land-cover types.

The June 2019 sowing also included a cover crop of barley that was absent from previous years, although the use of cover crops has been shown to increase soil microbial activity [48]. However, there we no observed differences related to barley addition (Figure 2), and in the 2019 (evaluation period) growing season there was a month-long difference between observed and simulated uptick of NEE and Reco (Figure 2b,c), which was not observed in the 2018 (calibration) season and requires some explanation. According to Khan, [49], tillage, which is the standout feature of the 2019 (evaluation) period, can stimulate soil microbial activity and thus respiration. Nevertheless, simulated NEE was not significantly different from observed (–14%), in line with the findings of Deng et al. [42] who also reported a good match for NEE, and of Abdalla et al. [27] who reported a corelation of R = 0.6 for NEE simulations compared to measured data on permanent grassland in the Republic of Ireland.

*4.2. Soil Climate*

Model performance in terms of simulating temperature at the soil surface (5 cm) was exceptional, which was perhaps to be expected given the version of the DNDC model we used was aimed specifically at improving surface exchange of energy fluxes and soil frost/thaw dynamic simulations [41,42], as is evident from the close correlation between simulated and measured outputs (Table 5). Soil moisture (WFPS) at 5cm followed a similar, though less striking trend and tended to underestimate by 11.2%, (Table 5). On the other hand, at 20 cm WFPS simulation quality was much lower, and although the two datasets matched closely the spread of the data was such that it was not possible to make a strong correlation. This discrepancy may be due to the model sensitivity to the soil water/ice status to changes in soil temperature when this was close to zero since a variation of $\pm 1 \,^{\circ}\text{C}$ above or below freezing is small for soil temperature but makes the difference between liquid water and ice in the soil.

*4.3. Biomass*

The DNDC model was able to simulate biomass production accurately for the first cut in the two-cut system used in Finnish pastures although the second cut tended to underestimate. Model performance in the first cut was closer to measured figures than in the second (2.8 ± 0.008% and 25.9 ± 0.05% respectively, Figure 4) and tended to assume higher growth rates after the first cut than were observed in the field. Testing across all years indicated that there was a significant difference between simulated and measured biomass, meaning that the model underperformed as a grass biomass prediction tool when calibrated to greenhouse gas fluxes.

## 5. Conclusions

This study demonstrated that the DNDC model is able to simulate GHG fluxes, soil climate conditions in a boreal grassland on a mineral soil within reasonable levels of accuracy, albeit at a trade-off in accuracy of crop biomass prediction. Future work using DNDC could be aimed at improving crop phenology (accounting for accurate onset and end of the growing seasons), interactions among plant species and potential benefits of legume crops in legume grassland systems, and improving the characterization of heat and water exchange at the soil surface layer to determine key factors influencing simulated GHG exchanges. Overall, however, our results suggest that the model is suitable for modelling crop, soil and GHG exchange from boreal grasslands.

**Author Contributions:** D.F.—data analysis, model calibration and evaluation, original draft. J.D.—model calibration and evaluation, editing and advisory. M.T.H.—review, editing and advisory. N.S.—review, editing, advisory and project supervision. All authors have read and agreed to the published version of the manuscript.

**Funding:** This work was supported by funding from the Ministry of Agriculture and Forestry Finland (Helsinki, FI) (Project: Clover for biogas, Project NC-GRASS: VN/28562/2020-MMM-2).

**Data Availability Statement:** The data presented in this study are available on request from the corresponding authors.

**Conflicts of Interest:** We declare no conflict of interest.

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
