# Peer review of "Simulating Soil-Plant-Climate Interactions and Greenhouse Gas Exchange in Boreal Grasslands Using the DNDC Model"

_land, doi:10.3390/land11111947_

Round 1
Reviewer 1 Report
The manuscript “Simulating soil-plant-climate interactions and greenhouse gas exchange in boreal grasslands using the DNDC model” by Forster et al. deals with application of the process-based model for modelling carbon exchange in managed boreal grassland ecosystem. The methods of the study are well described. Results are clear and reasonable. The manuscript results are important for the studying of carbon balance in boreal ecosystem.
I recommend to publish the manuscript after minor revision.
Specific comments
Figures 2-3 Should be improved to clarify differences between observed and modeled data. I suggest to add plots with Difference (OBS-MOD) vs MOD data.
Please discuss the largest differenced observed between field data and simulations for GPP and Reco in August 2018, Reco and NEE in June 2019.
DNDC model is capable to reproduce Nitrogen fluxes. Could you add some information about obtained simulated N fluxes?
Author Response
First of all, thankyou for taking the time to review our article. We have discussed your points and made some corrections and changes to reflect this and feel that this has improved the paper significantly. Please find the following replies to your points:
Figures 2-3 Should be improved to clarify differences between observed and modeled data. I suggest to add plots with Difference (OBS-MOD) vs MOD data.
These figures have been adapted for clarity, measured data is now a red line and simulated data is in circles. Adding OBS-MOD plots isn’t usual for these sorts of studies, and we felt this did not add value to the study.
Please discuss the largest differenced observed between field data and simulations for GPP and Reco in August 2018, Reco and NEE in June 2019.
Spikes in GPP and Reco in 2018 are probably related to increased rainfall at the same time (figure 1). The increases in Reco and NEE in 2019 are discussed in lines 269-276.
DNDC model is capable to reproduce Nitrogen fluxes. Could you add some information about obtained simulated N fluxes?
N specifically was not included as no measured N fluxes were available to draw such a comparison, so we regarded it as outside the scope of this study which is based on validation against eddy covariance data.
Reviewer 2 Report
This is an important study by a highly professional international team of researchers. The method description and the review of existing DNDC studies is very informative. Results are clear and well described and statistical tests are up-to-date. The authors are candid about limitations encountered in their study and make realistic suggestions for follow-up research. Excellent!
I have two side comments in addition: (1) we learn about grassland in Finland but in the introduction of the paper the important issue is raised as to the effect of the extension of arable land into high-latitude regions with virgin soils. We only learn about existing conditions in the Finland soil with grasses. Can anything be said about the conditions before this field was established? Some climate-change denyers state that efforts in areas with moderate climates are meaningless considering what will happen when virgin high-latitude soils will be cultivated as the amount of released methane would be overwhelming. Are data available to debunk that story for this particular location ( without permafrost it seems)? (2) why not refer to the UN Sustainable Development Goals to link the story more directly to the international policy and science debate? Here we deal with SDG2 and 13. .
Author Response
First of all, thankyou for taking the time to review our article. We have discussed your points and made some corrections and changes to reflect this and feel that this has improved the paper significantly. Please find the following replies to your points:
(1) we learn about grassland in Finland but in the introduction of the paper the important issue is raised as to the effect of the extension of arable land into high-latitude regions with virgin soils. We only learn about existing conditions in the Finland soil with grasses. Can anything be said about the conditions before this field was established? Some climate-change denyers state that efforts in areas with moderate climates are meaningless considering what will happen when virgin high-latitude soils will be cultivated as the amount of released methane would be overwhelming. Are data available to debunk that story for this particular location ( without permafrost it seems)?
You raise an interesting point. However, this study focusses on mineral soils (without permafrost) rather than organic ones, and methane emissions from such soils would be much less than if converting virgin peat soils. We therefore felt that we couldn’t justify comment on this topic.
(2) why not refer to the UN Sustainable Development Goals to link the story more directly to the international policy and science debate? Here we deal with SDG2 and 13.
Thankyou for bringing this to our attention. Mention has been made in paragraph 1 (lines 31-34).
Reviewer 3 Report
General Comments:
"The article provides a good overview about eddy-covariance, soil climate and biomass data to evaluate the simulation of GHG emissions from grassland in eastern Finland using the process-based model DNDC. It is within the scope of the journal, well written and well structured, with a quality production, at the level of your journal. There is an adequate number of bibliographic references (48), and in general they are recent studies, nothing very old. The objectives are clear and concise, the methodology is adequate for the proposal, the mathematical formulations for statistic are ok, the area of study was well defined. The simulations were well defined and the figures are of good quality. I did not identify methodological errors or open questions about the proposal. These results on the acceptable representativeness of model data to reproduce the carbon balance in the studied area are a significant scientific advance, due to the impossibility of having in situ measurements for many more points, in addition to being able to estimate the impacts of climate change on this balance. For this reason, I see that the main discussion that needs to be increased in the article is: 1- more explanations of the reason for the discrepancies in the GPP peaks in the years 2018 and 2019, because in 2017 this discrepancy was much smaller."
Author Response
First of all, thankyou for taking the time to review our article. We have discussed your point and made some corrections and changes to reflect this and feel that this has improved the paper significantly. Please find the following replies to your point:
1- more explanations of the reason for the discrepancies in the GPP peaks in the years 2018 and 2019, because in 2017 this discrepancy was much smaller."
These appear to be an artifact of increased rainfall occurring at these times (which are otherwise fairly dry periods) being interpreted by DNDC corresponding to increased GPP. This is commented on in lines 271-280.